# Cortex-dependent recovery of unassisted hindlimb locomotion after complete spinal cord injury in adult rats

Anitha Manohar[1†], Guglielmo Foffani[2,3†], Patrick D Ganzer[1], John R Bethea[4], Karen A Moxon[1,5*]

[1]School of Biomedical Engineering, Science and Health Systems, Drexel University, Philadelphia, United States; [2]CINAC, Hospital Universitario HM Puerta del Sur, Universidad CEU-San Pablo, Madrid, Spain; [3]Hospital Nacional de Parapléjicos, Servicio de Salud de Castilla-La Mancha, Toledo, Spain; [4]Department of Biology, Drexel University, Philadelphia, United States; [5]Department of Biomedical Engineering, University of California, Davis, United States

**Abstract** After paralyzing spinal cord injury the adult nervous system has little ability to 'heal' spinal connections, and it is assumed to be unable to develop extra-spinal recovery strategies to bypass the lesion. We challenge this assumption, showing that completely spinalized adult rats can recover unassisted hindlimb weight support and locomotion without explicit spinal transmission of motor commands through the lesion. This is achieved with combinations of pharmacological and physical therapies that maximize cortical reorganization, inducing an expansion of trunk motor cortex and forepaw sensory cortex into the deafferented hindlimb cortex, associated with sprouting of corticospinal axons. Lesioning the reorganized cortex reverses the recovery. Adult rats can thus develop a novel cortical sensorimotor circuit that bypasses the lesion, probably through biomechanical coupling, to partly recover unassisted hindlimb locomotion after complete spinal cord injury.

*For correspondence: moxon@ucdavis.edu

[†]These authors contributed equally to this work

**Competing interests:** The authors declare that no competing interests exist.

## Introduction

Spinal cord injury leads to dramatic loss of motor function below the level of the lesion, often permanently compromising the ability to locomote (*van Middendorp et al., 2011*). From a translational research perspective, recovery of motor function and locomotion can in principle be achieved with at least two conceptually different approaches: (a) 'healing' the lesion to re-establish motor pathways (*Richardson et al., 1980*; *David and Aguayo, 1981*); (b) bypassing the lesion to reconnect motor circuits above and below (*Ethier et al., 2012*; *Manohar et al., 2012*; *Bouton et al., 2016*). The first approach pursues the regeneration ideal of overcoming the natural limited ability of the adult spinal cord to restore connections through the injury (*Selzer, 2003*; *Liu et al., 2010b*; *Ruschel et al., 2015*). The second approach represents the practical alternative of extracting motor information at higher levels and reinserting it below the lesion, either within the spinal cord, directly to the muscles or through a robotic actuator (*Bensmaia and Miller, 2014*; *Moxon and Foffani, 2015*).

The two approaches – healing vs bypassing the lesion – implicitly assume that the adult nervous system is unable to innately develop a recovery strategy to bypass the lesion. Here we will challenge this assumption. Specifically: (i) we chose a spinal cord injury model – complete thoracic transection in adult rats – that excludes any spinal transmission of motor commands through the lesion; and (ii) we delivered a combination of pharmacological treatment (5-HT agonists) and physical therapies

(active treadmill training and/or passive hindlimb bike exercise) designed to maximize reorganization of sensorimotor circuits both below and above the level of the lesion (*Courtine et al., 2009*; *Kao et al., 2009*, *2011*; *Ganzer et al., 2013*; *Graziano et al., 2013*; *Oza and Giszter, 2014*, *2015*; *Foffani et al., 2016*; *Ganzer et al., 2016*).

We show that maximizing the reorganization of the hindlimb sensorimotor cortex creates a novel circuit that responds to input from the ventral forepaws and activates trunk muscles that span the lesion. This new cortical circuit bypasses the lesion, probably via biomechanical coupling, allowing animals to recover a surprising level of unassisted hindlimb weight support and locomotion after complete spinal cord injury.

## Results

### Combined therapeutic interventions induce hindlimb motor recovery after complete spinal cord injury

We first sought direct evidence that hindlimb motor recovery after spinal cord injury can indeed be achieved without explicit spinal transmission of motor commands through the lesion. To this end, in the first set of experiments adult rats (n = 45) received a complete spinal cord transection at thoracic level (T9/T10) and were treated with 5-HT agonists combined with either (a) physical therapy below the level of the lesion (passive hindlimb bike exercise, n = 15) or (b) physical therapy below and above the level of the lesion (passive hindlimb bike exercise + active treadmill training, n = 15), and were compared with a group of transected animals that received 'sham' therapy (n = 15). For simplicity, we will refer to 5-HT agonist + passive hindlimb bike exercise as 'partial therapy' and 5-HT agonists + passive hindlimb bike exercise + active treadmill training as 'complete therapy'. Hindlimb motor recovery was assessed both in controlled experimental conditions, as percentage of weight supported step cycles (%WSS) during treadmill locomotion with lateral but no vertical assist (at 4, 8 and 12 weeks post transection), and in more naturalistic conditions using open field testing without any support (at 2, 4, 8 and 12 weeks post transection, normalized to group-averages at week two post transection). Behavioral measures in all groups were always performed after acute administration of drugs (the same 5-HT agonists used for therapy), in order to guarantee the functional state of the cord below the level of the lesion (*Barbeau and Rossignol, 1990*; *Jackson and White, 1990*), but without any sensory stimulation (i.e. tail pinch or perineal stimulation).

These combined therapeutic interventions markedly improved hindlimb motor performance as measured by %WSS during treadmill locomotion (2-way mixed ANOVA, time x therapy: $F_{(4,78)}=3.7$, p=0.0082; *Figure 1A*). As expected, 4 weeks after transection the %WSS was low for all groups (1.6 ± 4.3%). Conversely, the %WSS increased by a factor of ten at 8 weeks and 12 weeks compared to 4 weeks both, for animals receiving partial (Tukey: p=0.0644, p=0.0086) and complete therapy (p=0.0300, p=0.0002), but not for transected animals receiving sham therapy (p>0.99). Moreover, at 12 weeks, the %WSS of animals receiving complete therapy (19.4 ± 17.8%) was significantly higher than that of animals that received sham therapy (1.6 ± 2.9%, p=0.0103) but animals that received partial therapy were not different from sham therapy animals (12.3 ± 15.4%, p=0.32), suggesting that complete therapy was more effective than partial therapy. Many of the WSS achieved with either therapy were consecutive (*Figure 1A*, inset). After 12 weeks of complete therapy, three animals were able to perform >40% of WSS (one animal went as high as 56%) during the treadmill session.

Even though %WSS were always evaluated without any assisted vertical weight support, we also evaluated the minimal amount of assisted vertical weight support that was necessary for animals to maintain a consistent quadrupedal locomotion on the treadmill, defined as the load-bearing failure point (*Timoszyk et al., 2005*). The assisted vertical weight support at load-bearing failure point markedly decreased at 8 weeks and 12 weeks with partial and complete therapy but not with sham therapy (*Figure 1—figure supplement 1*), providing additional direct evidence of increasing levels of hindlimb weight support achieved by the animals.

These important motor recoveries were confirmed by the evaluations in the open field without any support (*Figure 1B*). Specifically, open field scores (*Basso et al., 1995*) were similar across groups at 2 weeks (6.5 ± 1.3 for sham, 5.7 ± 2.0 for partial, 5.2 ± 2.4 for full; 1-way ANOVA, $F_{(2,41)}$ =1.7, p=0.19), and then increased at 4 weeks, 8 weeks and 12 weeks compared to 2 weeks for

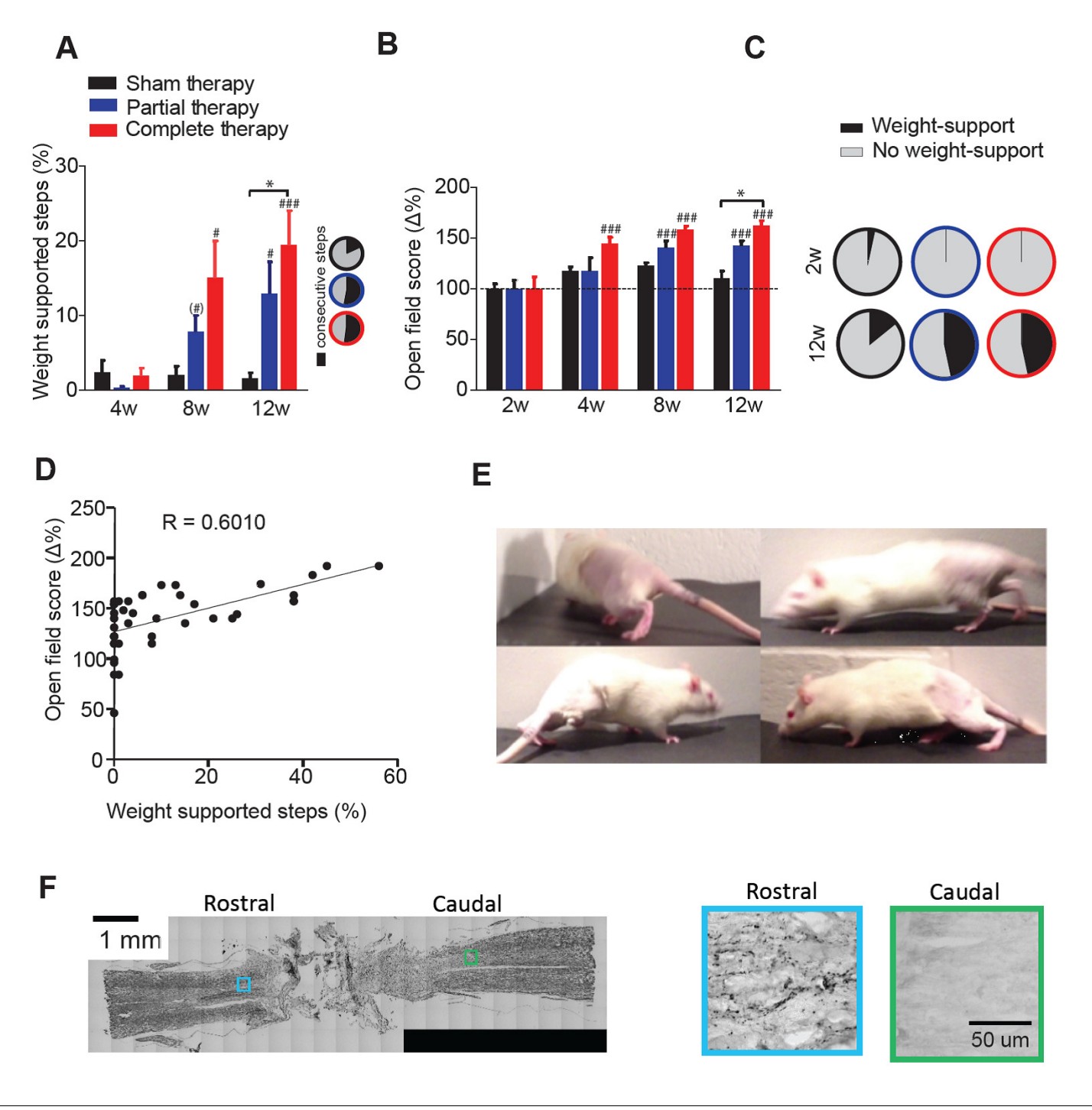

**Figure 1.** Locomotor recovery after a complete spinal cord injury. (**A**) Percentage of weight supported step cycles (% WSS) during treadmill locomotion for complete therapy (red), partial therapy (blue) and sham therapy (black) groups at 4, 8 and 12 weeks post-injury. Pie charts indicating the fraction of these weight supported steps that were part of a consecutive bout of three or more step cycles for each group (right panel inset, black: percentage of consecutive steps, grey: nonconsecutive steps). (**B**) Open field score measured by the BBB scale, expressed as % change from baseline at 2 weeks post injury. (**C**) Pie charts showing the proportion of BBB scores that correspond to weight support in the hindlimbs during unassisted open field locomotion (≥9) at 2 weeks (top panel) and 12 weeks (bottom panel). (**D**) Correlation between the open field score and %WSS at 12 weeks after injury. (**E**) Example frames of a spinalized rat after 12 weeks of complete therapy taking plantar weight supported steps with its hindlimbs in the open field. (**F**) Histological verification of complete spinal transection using Nissl stain (left) of a horizontal section of the spinal cord. 5-HT stain (right) of a rostral and caudal location taken from a slice adjacent to the Nissl stained slice on the left. Boxes represent the approximate location of the higher magnification picture

*Figure 1 continued on next page*

*Figure 1 continued*

of 5-HT stain. # represents differences from week 2 or 4 and * represents differences within the same week. (*) p<0.1, *p<0.05, **p<0.01, ***p<0.001. Error bars indicate 95% confidence intervals.

The following figure supplements are available for figure 1:

**Figure supplement 1.** Assisted weight support provided during training.

**Figure supplement 2.** A schematic representation of the body weight support device that was used during treadmill training.

animals receiving complete therapy (time x therapy: $F_{(6,123)}$=3.2, p=0.0064; Tukey: p<0.0001) and increased at 8 weeks and 12 weeks compared to 2 weeks for animals receiving partial therapy (p<0.0007), but not for transected animals receiving sham therapy (p>0.40). Again, at 12 weeks the open field scores of animals that received complete therapy were significantly higher than those of animals that received sham therapy (p=0.0137) but animals that received partial therapy were not different from sham (p=0.43), confirming that complete therapy was more effective than partial therapy.

To gain insight into the behavioral significance of the recovery in the open field, we focused on a critical aspect of motor performance: the ability of the animals to gain plantar weight support in the hindlimbs (BBB $\geq$ 9). The number of animals that achieved plantar weight support in the open field dramatically increased with partial or complete therapy from <4% at 2 weeks to 47% at 12 weeks, compared to 14% at 12 weeks in animals undergoing sham therapy (*Figure 1C*).

Interestingly, 12 weeks after spinal transection motor performance during weight-supported treadmill locomotion and open-field motor recovery were highly correlated (Pearson: R = 0.60, p<0.0001; n = 42; *Figure 1D*), suggesting that these two measures – quantitative on a treadmill with lateral support and semi-quantitative in the open field – are capturing similar aspects of functional recovery.

Taken together, these results show that even with a complete transection of the spinal cord, carefully designed combinations of therapeutic interventions can restore a significant level of hindlimb weight support and locomotion without any sensory stimulation or assisted vertical weight support after complete spinal cord injury (*Figure 1E*). Importantly, the completeness of the spinal lesion was histologically confirmed in all animals with both nissl and 5-HT staining (*Figure 1F*)

## Combined therapeutic interventions induce reorganization of the deafferented motor cortex after complete spinal cord transection

A critical aspect for understanding the possible mechanisms underlying the observed motor recovery is the role of cortical reorganization. At the end of the study, 12 weeks after the spinal transection, a subset of animals (n = 36) underwent an anesthetized motor mapping procedure using intracortical microstimulation to assess the possible reorganization of the deafferented hindlimb motor cortex under the different therapy regimens (complete therapy: n = 14, partial therapy: n = 12, sham therapy: n = 10). Motor reorganization was assessed in terms of cortical area (in mm$^2$) in which microstimulation elicited movement of the hindlimbs, trunk, forelimbs or vibrissae. An additional group of naïve intact animals (n = 5) was used to confirm the coordinates of hindpaw motor representation.

As expected, motor representations of the hindlimbs – typically observed in naïve intact rats – were absent in all lesioned animals, confirming the completeness of the spinal transection (*Figure 2A*). However, there was a significant therapy-dependent reorganization of the deafferented motor cortex (2-way mixed ANOVA, location x therapy: $F_{(4,66)}$=3.6, p=0.0103; *Figure 2A,B*). Specifically, we observed an expansion of the trunk representation in animals that received complete therapy compared to animals that received partial therapy (Tukey, p=0.0484) or sham therapy (p=0.0013). This expansion was not secondary to an in increase in cortical excitability, since there were no differences between the threshold currents across groups (*Figure 2—figure supplement 1*). We did not observe any expansion of the cortical representations of the forelimb or of the vibrissae into the deafferented motor cortex (p>0.99).

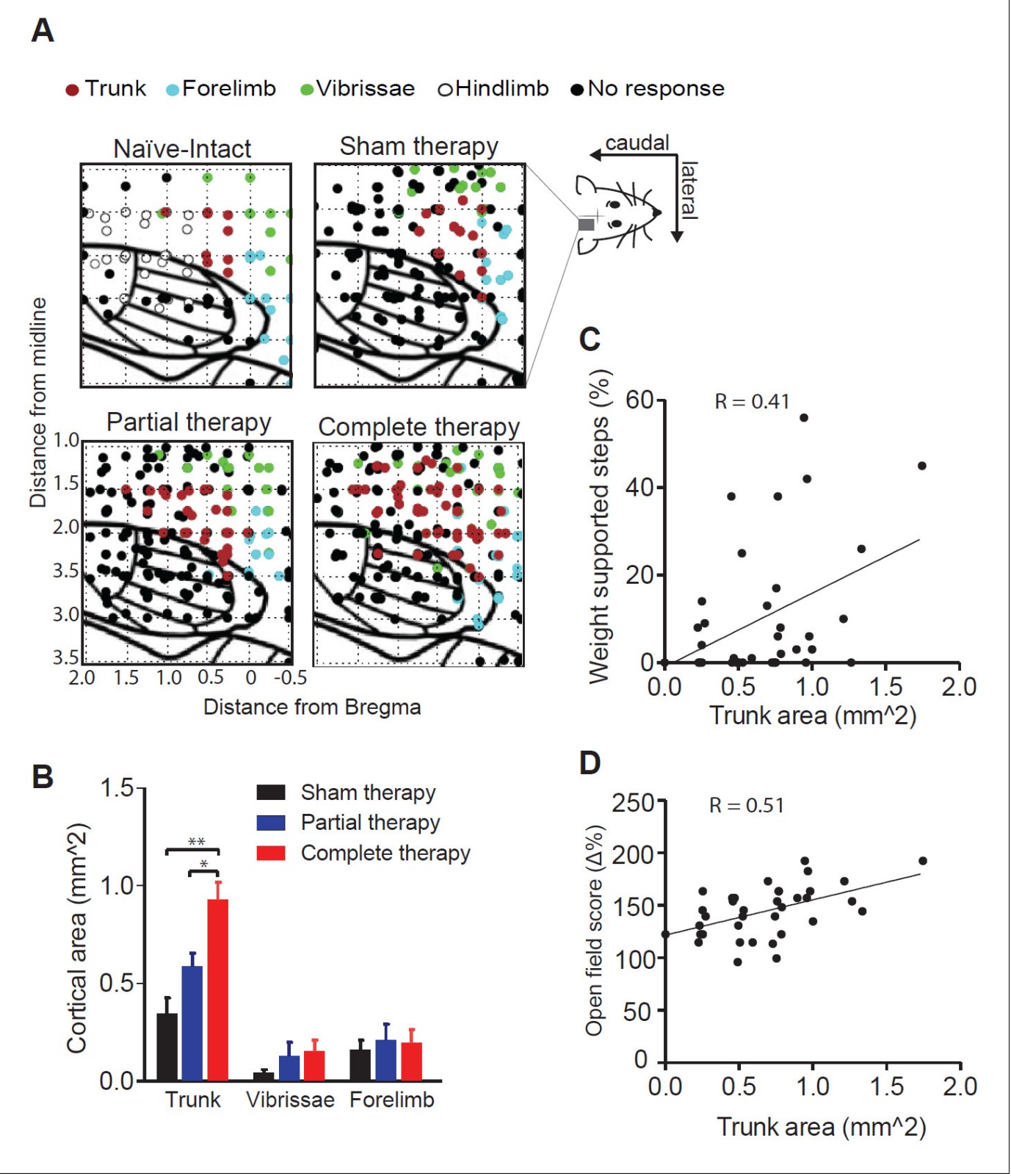

**Figure 2.** Reorganization of the motor cortex induced by combined therapeutic interventions. (**A**) Raw intra-cortical micro-stimulation motor maps showing the penetrations on a cortical grid (co-ordinates AP: from 0.5 mm rostral to bregma to 2 mm posterior to bregma; ML: 1 to 3.5 mm lateral to midline) color coded by the type of movement evoked for naïve-intact, sham therapy, partial therapy and complete therapy groups. (**B**) Average cortical area (mm²) corresponding to a specific type of movement (trunk, forelimbs or vibrissae) for all therapy groups. Correlation between cortical area

*Figure 2 continued on next page*

*Figure 2 continued*

corresponding to trunk movements and locomotor recovery measured by (C) % WSS and also (D) open field score. *p<0.05, **p<0.01. Error bars indicate 95% confidence intervals.

The following figure supplements are available for figure 2:

**Figure supplement 1.** Threshold currents during intra-cortical micro-stimulation motor maps.

**Figure supplement 2.** Correlation measures.

Importantly, considering all animals, the cortical area of the trunk representation correlated with motor performance, measured by both %WSS (Pearson: R = 0.41, p=0.0142; n = 36; *Figure 2C*) and open field scores (Pearson: R = 0.51, p=0.0013; n = 36; *Figure 2D*; *Figure 2—figure supplement 2*).

## Lesioning the reorganized motor cortex reverses the behavioral recovery

A correlation between cortical reorganization and motor recovery does not necessarily imply a causal relationship between the two. In order to find more convincing evidence that the reorganization of the deafferented hindlimb motor cortex directly contributed to the motor recovery, at the end of the motor map the reorganized motor cortex was electrolytically lesioned bilaterally (*Figure 3A*), animals received two additional weeks of therapy (partial or complete) post lesion and were then

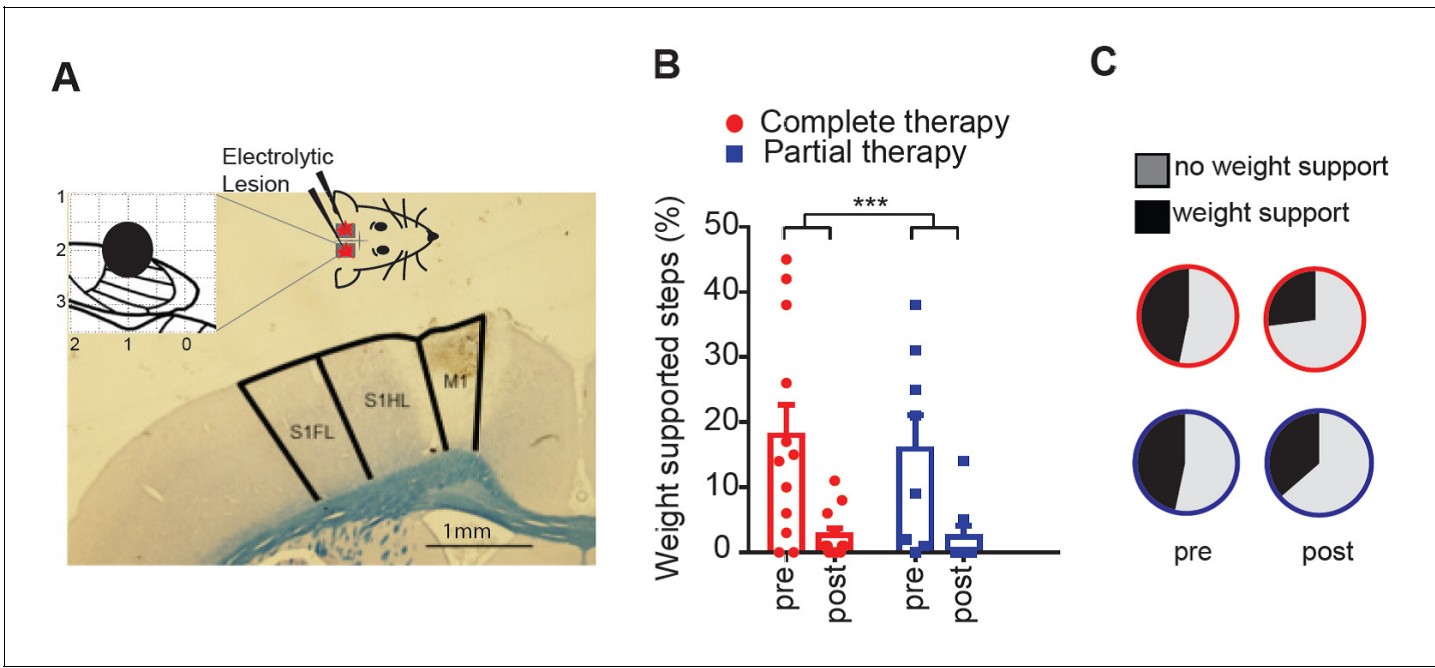

**Figure 3.** Lesioning the motor cortex reverses behavioral recovery. (A) Photomicrograph of exemplar electrolytic lesion in the hindlimb motor cortex (inset: cortical map showing lesion location centered at 1.0 mm caudal to bregma and 2.0 mm lateral). (B) Locomotor recovery measured by %WSS during treadmill locomotion evaluated before and after (pre-, post-) lesioning the motor cortex. (C) Proportion of BBB scores that correspond to weight support in the hindlimbs during unassisted open field locomotion (≥9) before and after (pre-, post-) cortical lesions. ***p<0.001. Error bars indicate 95% confidence intervals.

The following figure supplement is available for figure 3:

**Figure supplement 1.** Quality of steps assessed from the videos based on individual paw placements.

behaviorally re-evaluated at week 14 post transection (complete therapy, n = 12; partial therapy, n = 10).

Lesioning the reorganized motor cortex dramatically decreased the motor performance as measured by %WSS during treadmill locomotion (2-way ANOVA, lesion: F(1,20)=24.0, p<0.0001; lesion x therapy: F(1,20)=0.1, p=0.70; *Figure 3B*). The key element that differentiated animals that achieved good recovery from animals that displayed poor recovery was the ability to make plantar contact with the treadmill during stepping, which was lost after the cortical lesion (*Figure 3—figure supplement 1*). In agreement, the cortical lesion also decreased the number of animals that achieved plantar weight support in the open field (BBB ≥ 9) from 47% at week-12, to 36% and 27% after the cortical lesion in animals under partial or complete therapy, respectively (*Figure 3C*).

The %WSS for the forelimbs was 100% for all animals at week-12 and remained 100% for all animals at week-14, after the cortical lesion. Furthermore, 100% of forelimb steps were plantar steps both before and after the cortical lesion. Forelimb weight support thus remained unobstructed by the cortical lesion, supporting the specificity of the cortical lesion to hindlimb function.

As a control, the same cortical lesion performed in a group of naïve intact animals (n = 4) did not induce any change in motor performance either on the treadmill or in the open field (100%WSS and BBB = 21 both before and after the cortical lesion).

These results suggest that the reorganization of the motor cortex is at least partly responsible for the hindlimb motor recovery after complete spinal cord transection.

## Corticospinal fibers from the reorganized motor cortex sprout into the thoracic spinal cord after therapy

The behavioral recovery induced by therapeutic interventions could be mediated by the sprouting of corticospinal fibers within the spinal cord. To test this possibility, in a second set of experiments, we labeled the projections from the reorganized motor cortex to the spinal cord with injections of the anterograde tracer BDA (*Figure 4A*), in transected animals receiving sham therapy (n = 2) and transected animals receiving complete therapy (n = 2). A group of naïve intact animals (n = 2) was used as a control. The tracer was injected in the deafferented hindlimb motor cortex (i.e. the reorganized cortex) 9 weeks after the spinal transection, animals continued therapy and were perfused 3 weeks later. For each animal, labeled axons were counted in the gray matter of the spinal hemicord at the thoracic level where alpha motor neurons innervate trunk musculature. Specifically, axons were counted at each of three different levels above the lesion (T1, T4 and T7) contralateral to the injected cortex, from five representative slices spaced 250 microns apart (*Figure 4B*). Average counts per slice were considered as independent samples.

There was a rostro-caudal gradient of axonal innervation from the hindlimb motor cortex into the spinal cord (two-way ANOVA, level: F(2,81)=7.3, p=0.0012), with more counts at T7 level compared to T4 (Tukey, p=0.0054) and T1 (p=0.0030). More importantly, therapy after spinal cord transection promoted corticospinal sprouting (therapy, F(2,81)=6.4, p=0.0027). Specifically, more axons were counted in the gray matter of the thoracic spinal cord in transected animals that underwent complete therapy compared to both transected animals that underwent sham therapy (p=0.0111) and naïve intact animals (p=0.0055) (*Figure 4C–F*).

These results show that our therapy regimen promotes sprouting of corticospinal fibers, originating in the deafferented hindlimb cortex, into the gray matter of the spinal cord above the level of the lesion. This corticospinal sprouting could allow animals to gain greater control of trunk musculature, thus contributing to the observed motor recovery.

## Combined therapeutic interventions induce reorganization of the deafferented somatosensory cortex after complete spinal cord transection

Recovery of hindlimb function after spinal cord transection likely relies on somatosensory inputs from the intact forelimbs, suggesting that motor cortical reorganization might be paralleled by somatosensory cortical reorganization in our animals. To test this prediction, in a third set of experiments we chronically and bilaterally implanted 16-channel arrays of microelectrodes (4-by-4) in the hindlimb somatosensory cortex of 14 animals, which were then spinally transected and received either complete therapy (n = 8) or sham therapy (n = 6). We mapped the responses of cortical

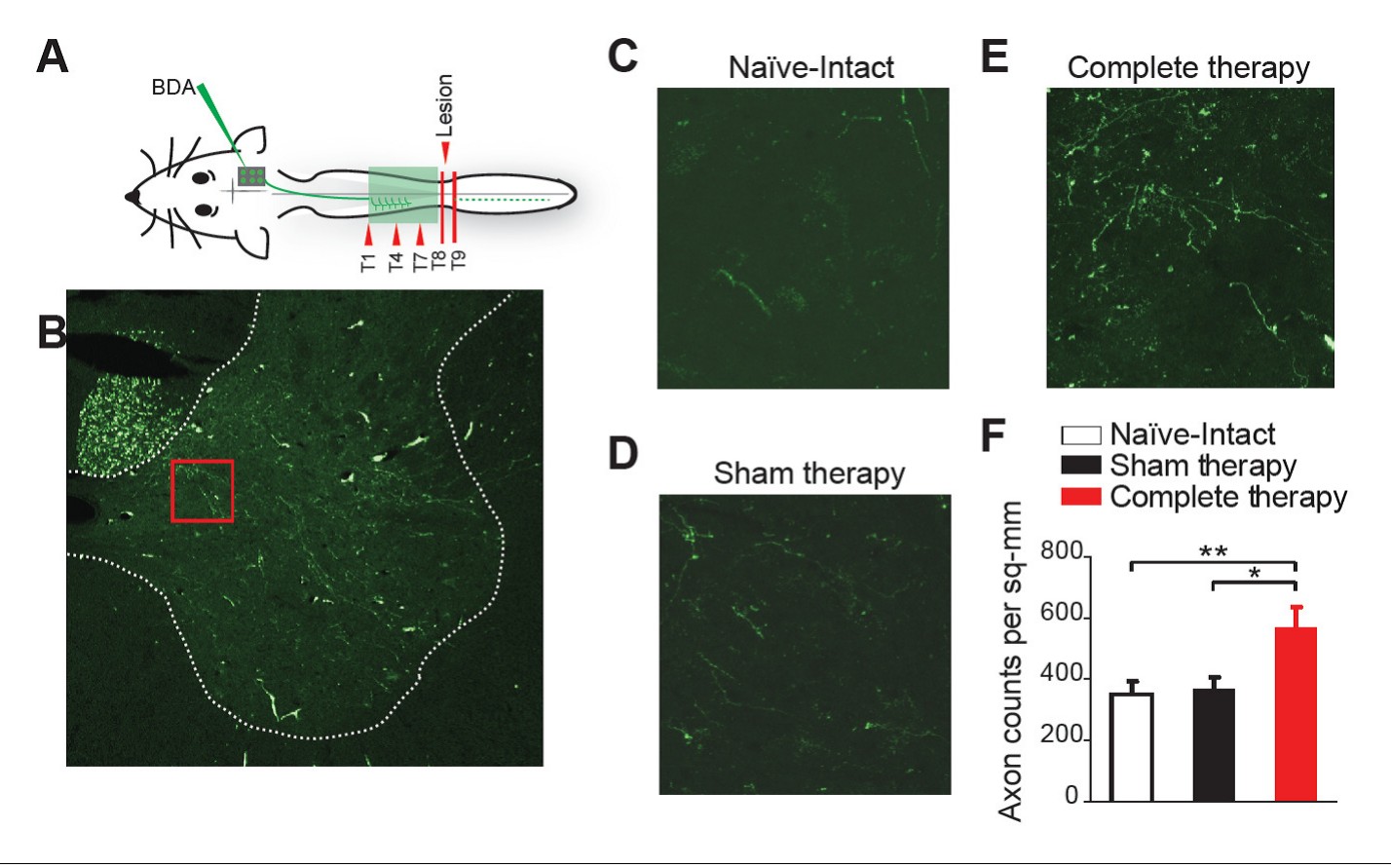

**Figure 4.** Corticospinal sprouting into the thoracic spinal cord. (A) Diagram illustrating anterograde tracing experiments. (B) Histology of a sample coronal slice of the thoracic spinal hemi-cord taken from above the level of the spinal lesion and a magnified view of the sprouting into the grey matter in (C) Naïve-Intact (D) Sham therapy and (E) Complete therapy groups. (F) Bar graphs showing the average count of corticospinal axons sprouting per mm2 of gray matter of the spinal hemi-cord for the different groups. *p<0.05, **p<0.01. Error bars indicate 95% confidence intervals.

neurons in the deafferented hindlimb cortex to tactile sensory stimuli delivered to seven fixed locations on each forelimb, pre (week 0) and post spinal cord injury (week 4, 8, 12). Cortical reorganization was measured as the percentage responding of cells, i.e. cells recorded in the deafferented hindlimb cortex that significantly responded to tactile stimulation of at least one location on the contralateral forelimb (*Figure 5A,B*). Note that neurons in the hindlimb cortex did not have significant responses to trunk stimulation either before or after spinal cord injury, likely because the trunk sensory area is not topographically adjacent to the deafferented hindlimb region.

We recorded an average of $33.4 \pm 9.7$ neurons per day per animal, with no differences in the number of neurons recorded between animals that received complete or sham therapy (two-way ANOVA, therapy: $F(1,43)=0.1$, $p=0.77$) or across evaluation days (time: $F(3,43)=0.3$, $p=0.79$). However, the percentage of responding cells was highly dependent on whether transected animals received complete or sham therapy (two-way GZLM, therapy x time, Wald $\chi^2(3)=12.7$, $p=0.0054$). Namely, even though the two groups were expectedly not different at week zero before the spinal transection (Tukey, $p=0.42$), the percentage of responding cells became significantly higher in animals that received complete therapy compared to sham therapy at week 4 ($p=0.0128$), week 8 ($p<0.0001$) and week 12 ($p<0.0001$) after spinal cord transection (*Figure 5B*).

We then exploited the 4-by-4-matrix arrangement of our electrode arrays to test whether this increased responsiveness of the hindlimb somatosensory cortex to forelimb stimuli, induced by therapy, followed a specific spatial topography. We found that in animals that received complete therapy after spinal cord transection, the probability to record responsive cells expanded toward more medial and more rostral electrodes (*Figure 5C*). In fact, the response magnitude (spikes/stimulus) of

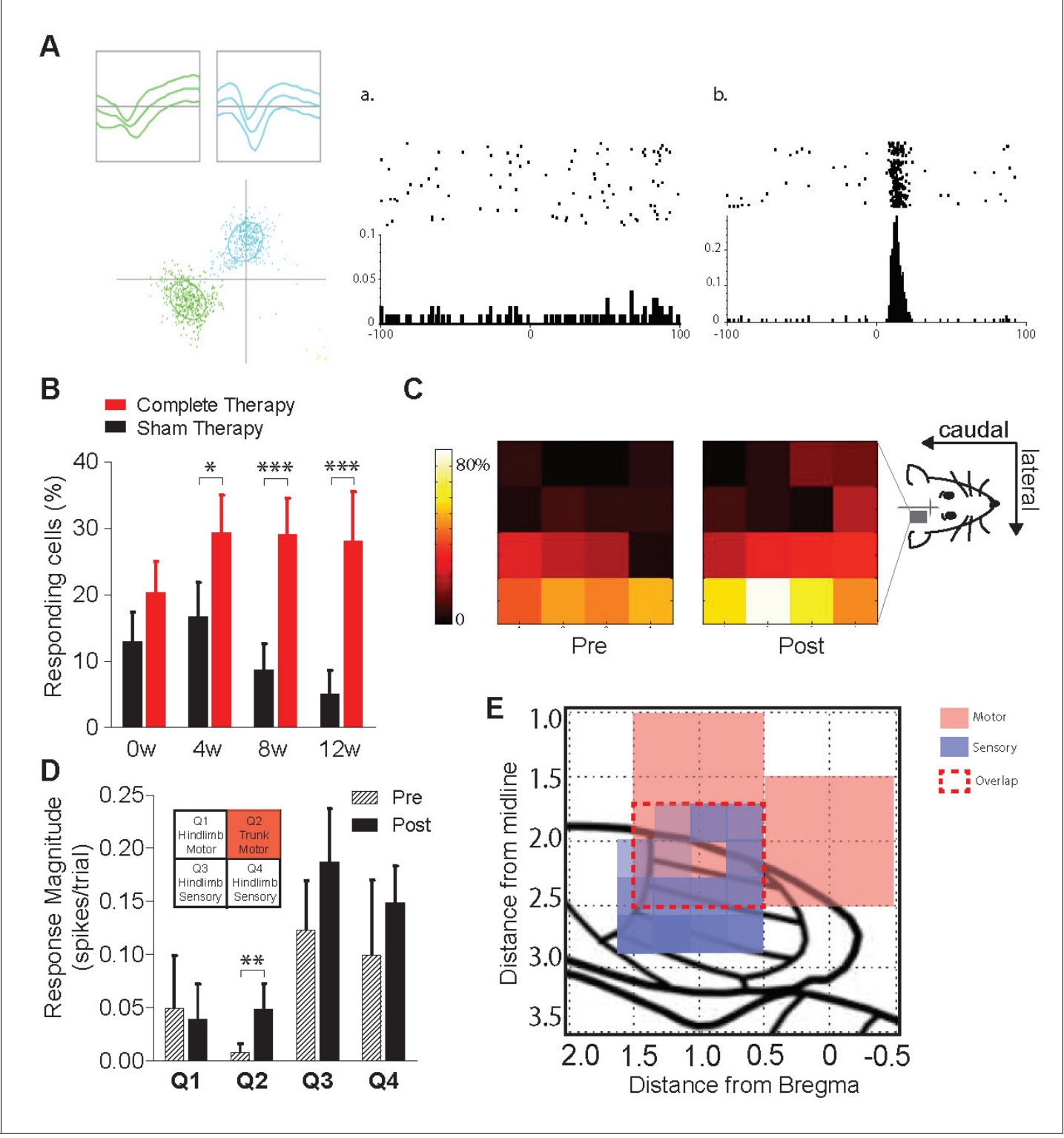

**Figure 5.** Reorganization of the hindlimb sensory cortex induced by combined therapeutic interventions. (**A**) Example showing single unit discrimination using waveform-based cluster analysis (left panel) and exemplar peri-stimulus time rasters and histograms of (**a**) a neuron which is unresponsive to sensory stimulation and (**b**) a neuron showing a significant response to sensory stimulation measured in a window extending 100 ms before and after the stimulus. (**B**) Proportion of neurons recorded from sham therapy group (black) that had a significant response to forepaw sensory stimulation compared to that of complete therapy group (red) pre- transection (0w) and weeks 4 (4w), 8 (8w) and 12 (12w) post-transection. (**C**) Heat maps indicating distribution of forepaw sensory responses recorded on each of the wires of the 4 × 4 microwire electrode array implanted into the hindlimb

*Figure 5 continued on next page*

*Figure 5 continued*

sensorimotor cortex pre- (week 0) and post- (weeks 4, 8, 12) spinal transection. (D) Average magnitude of the neural responses to forepaw stimulation broken down into four quadrants based on the location of the electrode wire (top left inset: quadrant locations), pre- and post- spinal transection. (E) Depiction of the rat sensorimotor cortex after complete spinal transection followed by 12 weeks of combined therapies. X-axis is the rostrocaudal distance from bregma, y-axis is the mediolateral distance from bregma. Red represents expansion of motor cortex, purple represents expansion of forelimb somatosensory cortex into the deafferented hindlimb cortex. Red dashed lines represent the extent of sensorimotor overlap within the confines of our recorded regions. *p<0.05, **p<0.01, ***p<0.001. Error bars indicate 95% confidence intervals.

The following figure supplement is available for figure 5:

**Figure supplement 1.** Characterization of neuronal responses to forepaw placement on the treadmill during locomotion.

the recorded cells specifically increased in the medial-rostral quadrant of the array (Q2; t-test, p=0.002; bonferroni-corrected alpha = 0.0125) but not in the other quadrants (p>0.118; *Figure 5D*).

This medial-rostral expansion of the forelimb somatosensory representation nicely overlaps with the expanded trunk motor representation we described above, suggesting that integrated sensorimotor cortical reorganization is likely to contribute to the motor recovery induced by therapy after spinal cord transection (*Figure 5E*). Importantly, neurons in the hindlimb cortex were responsive to forepaw placement during normal treadmill locomotion, with their peak response latency being distributed across a broad range of phases in the locomotion cycle (*Figure 5—figure supplement 1*). This broad normal responsiveness provides a possible functional substrate for the cortical reorganization after spinal cord injury

## Discussion

The main result of the present work is that completely spinalized adult rats can recover unassisted weight-supported hindlimb stepping without explicit spinal transmission of motor commands through the lesion. This apparently counterintuitive locomotor recovery was achieved with combinations of therapies that lead to substantial reorganization of the motor cortex, including significant expansion of the trunk representation into deafferented hindlimb cortex. This cortical reorganization is responsible for the motor recovery, as supported both by indirect correlative evidence and more directly by the observation that lesioning the reorganized cortex reverses the motor recovery (experiment 1; *Figures 1–3*). We further show that this cortical reorganization is associated with sprouting of cortico-spinal axons within the cord above the level of the lesion (experiment 2; *Figure 4*) and expansion of intact forelimb somatosensory representation toward the reorganized hindlimb motor cortex (experiment 3; *Figure 5*). These results suggest the creation of a novel cortical sensorimotor circuit that responds to inputs from the ventral forepaws and activates trunk muscles that likely span the lesion, causally subserving the observed behavioral recovery via putative biomechanical coupling. The adult nervous system is thus able to develop a strategy to bypass the lesion and enable – at least partial – recovery of hindlimb locomotion after complete spinal cord injury.

### How can weight-supported hindlimb stepping be recovered after complete spinal cord injury?

Several previous studies were able to improve hindlimb locomotion after complete spinal cord injury in adult rats, either using sensorimotor rehabilitation(*de Leon et al., 2002*; *Alluin et al., 2015*), administration of serotonergic agonists (*Feraboli-Lohnherr et al., 1999*; *Antri et al., 2002*, *Antri et al., 2005*), transplants below the level of the lesion (*Giménez Y Ribotta et al., 2000*, *Sławińska et al., 2013*) or combined pharmacological and electrical stimulation (*Gerasimenko et al., 2007*; *Ichiyama et al., 2008*; *Courtine et al., 2009*; *Musienko et al., 2011*). However, all previous interventions required either tail pinching, perineal stimulation and/or some level of assisted vertical weight support to achieve hindlimb locomotion in adult spinalized rats. In our study, during treadmill testing and open-field evaluation our animals received no form of peripheral stimulation or assisted vertical weight support.

At first glance, this recovery of weight-supported hindlimb stepping after complete spinal cord transection seems paradoxical. Apart from improbable regeneration across the lesion, the

immediate most logical explanation is to think about residual connections (*Waxman, 1989*; *Courtine et al., 2008*; *Cowley et al., 2010*; *Flynn et al., 2011*; *Zaporozhets et al., 2011*; *Shah et al., 2013*; *Filli et al., 2014*), but this possibility is excluded by both physiological and anatomical evidence in our experiments: (a) no movement whatsoever could be elicited below the level of the lesion by microstimulation of the hindlimb motor cortex; (b) the completeness of the spinal lesion was verified by both nissl and the more sensitive 5-HT staining in all animals at the end of the study.

An alternative explanation is suggested by the functional expansion of the trunk cortical representation. Trunk muscles do extend from above to below the level of the injury, thus providing a biomechanical substrate for bypassing the lesion. The idea is that our treated animals learned to activate the trunk musculature to reduce the load on the hindlimbs while balancing on the forelimbs, which allowed the central-pattern-generators below the level of the lesion to be activated, resulting in weight-supported hindlimb stepping. This explanation is supported by previous observations in neonatally spinalized rats (*Giszter et al., 2007*, *2008*, *2010*), which can develop unassisted weight-supported stepping (*Stelzner et al., 1975*; *Weber and Stelzner, 1977*; *Commissiong and Sauve, 1993*; *Kim et al., 2001*). Importantly, the motor recovery of neonatally spinalized rats is not due to any reconnection within the spinal cord (*Cummings et al., 1981*; *Tillakaratne et al., 2010*), but is instead causally related to sensorimotor cortical reorganization (*Giszter et al., 1998*, *2008*; *Kao et al., 2009*, *2011*; *Moxon et al., 2013*; *Oza and Giszter, 2015*; *Udoekwere et al., 2016*). Until now, recovery of unassisted weight-supported hindlimb stepping after complete spinal cord injury was believed to remain restricted to neonatally spinalized rats, due to the higher cortical plasticity at neonatal age compared to adulthood (*Oza and Giszter, 2015*; *Udoekwere et al., 2016*). Our results show that carefully designed combinations of pharmaceutical and physical therapies after spinal cord transection can produce sufficient cortical plasticity to recover unassisted weight supported hindlimb stepping after complete spinal cord injury in adult rats.

## Cortical plasticity after spinal cord injury

Plasticity can refer to neural reorganization for at least two levels: (i) structural changes at the synaptic-to-cellular level, or (ii) changes in receptive field properties of neurons at the cellular-to-population level (*Buonomano and Merzenich, 1998*; *Moxon and Foffani, 2015*). Our results clarify how cortical plasticity after spinal cord injury encompasses both levels.

Cellular-to-population plasticity is demonstrated in our experiments by the expansion of both the trunk motor representation and the forelimb somatosensory representation into the deafferented hindlimb cortex. This overlap of sensorimotor reorganization unifies previous studies on somatosensory (*Endo et al., 2007*; *Kao et al., 2009*; *Ghosh et al., 2010*; *Ganzer et al., 2013*; *Graziano et al., 2013*; *Humanes-Valera et al., 2017*) or motor (*Fouad et al., 2001*; *Giszter et al., 2008*; *van den Brand et al., 2012*; *Oza and Giszter, 2014*, *2015*; *Ganzer et al., 2016*) reorganization into a joint framework, and is likely to be a key factor for the observed functional recovery. Indeed, lesioning the reorganized cortex reversed the recovery of hindlimb function, without affecting forelimb function. Intriguingly, our cortical lesion did not produce any evident functional impairment in naïve non-spinal-cord-injured animals, in agreement with recent data suggesting that motor cortex is required for learning but not for executing a learned motor skill in normal rats (*Kawai et al., 2015*) and with the view that in rodents the motor cortex is not required for normal locomotion (*Courtine et al., 2007*). Crucially, our data suggest that the motor cortex is indeed required to sustain the recovered locomotion after complete spinal cord injury.

We admittedly did not perform motor and somatosensory maps in the same animals, nor did we attempt to selectively deactivate motor or somatosensory circuits, so we could not disentangle motor vs. sensory contributions of cortical reorganization to the observed recovery. This disentanglement might be important to understand why some animals achieved weight-supported stepping and other did not, which currently remains unclear. In any case, the relationship between cortical plasticity and functional recovery in our adult spinalized rats is very similar to neonatally spinalized rats, suggesting that combinations of therapies after spinal cord injury might be able to rescue at least some features of critical period plasticity (*Pizzorusso et al., 2002*; *Maya Vetencourt et al., 2008*; *Murphy and Corbett, 2009*; *Yamahachi et al., 2009*).

Synaptic-to-cellular plasticity is demonstrated in our experiments by the sprouting of cortico-spinal axons within the cord above the level of the lesion. This cortico-spinal sprouting is in agreement

with many previous studies in several models of spinal cord injury (*Fouad et al., 2001*; *Weidner et al., 2001*; *Bareyre et al., 2004*; *Barritt et al., 2006*; *Vavrek et al., 2006*; *Girgis et al., 2007*; *García-Alías et al., 2009*; *Sasaki et al., 2009*; *Ghosh et al., 2010*; *Lee et al., 2010*; *Wang et al., 2011*; *Floriddia et al., 2012*; *Scali et al., 2013*; *Du et al., 2015*). Future work should expand on the molecular mechanisms of this cortical plasticity, establishing the role of pre-synaptic and post-synaptic processes as well as the possible contribution of astrocytes and neuron-glia interaction in the cortical reorganization observed here. Overall, even though subcortical plasticity is likely to have contributed to the observed functional recovery (*Kambi et al., 2014*; *Alonso-Calviño et al., 2016*), our results suggest that at least part of the plasticity induced by our therapies after spinal cord injury is genuinely cortical.

## Pathophysiological implications

Our main result that cortical plasticity, induced in adult spinalized rats, supports recovery of weight-supported hindlimb stepping has several pathophysiological implications.

First, it is important to acknowledge the potential translational limitation of our study focusing on quadrupedal instead of bipedal stepping. Indeed, the novel cortical sensorimotor circuit developed in our rats – integrating sensory information from the ventral forepaws and motor control to the muscles of the spine – suggests that the closed-loop control of trunk musculature that is sufficient for quadrupedal functional improvement might be irrelevant for bipedal locomotion. However, it can be argued that patients attempting to recover locomotion after a spinal cord injury make significant use of the upper limbs to handle their walking aids, so that locomotion behavior switches from the normal bipedal pattern to an assisted quasi-quadrupedal pattern (*Del-Ama et al., 2014*). From this quasi-quadrupedal perspective, this novel supraspinal control of trunk stability – with possible involvement of intersegmental reflexes (*Tani et al., 1997*) – might become critical to achieve locomotion in recovering patients with spinal cord injury, as in our animals.

Second, from a reductionist perspective, it is tempting – and indeed valuable – to investigate the potential functional impact of different therapies delivered one at a time. However, therapeutic interventions after spinal cord injury can act at multiple levels of the sensorimotor system (*Onifer et al., 2011*; *Musienko et al., 2012*; *van den Brand et al., 2012*), including the skeletomuscular system (*Hangartner et al., 1994*; *Lauer et al., 2011*), muscular-spinal reflex circuits (*Mello et al., 2004*; *Hamid and Hayek, 2008*; *Phadke et al., 2009*; *Rayegani et al., 2011*), and neural circuits within the spinal cord (*Liu et al., 2010a*; *Côté et al., 2011*; *Keeler et al., 2012*). Consequently, when combined together, different therapies can have redundant or synergistic effects on cortical plasticity and recovery (*Foffani et al., 2016*). The present results support the view that well-designed combinations of therapies should be employed to maximize corticospinal plasticity and functional outcome (*Thuret et al., 2006*; *Hollis et al., 2016*).

Finally, our study substantially raises the standard of behavioral recovery that adult rats can achieve after a spinal cord injury without any spinal transmission of signals through the lesioned cord (and without any external support or sensory stimulation). Any work attempting to restore sensorimotor communication through the lesion after spinal cord injury should carefully consider the possibility that at least part of the functional recovery might be unrelated to the restored spinal communication.

## Conclusion

Overall, our results show that careful combinations of pharmacological and physical therapies in adult rats can create a novel cortical sensorimotor circuit that is able to bypass the lesion – probably through biomechanical coupling – to partly recover unassisted hindlimb locomotion after complete spinal cord injury. These results demonstrate the importance of taking advantage of plasticity along the entire neural axis when developing therapies to optimize recovery of function after severe spinal cord injury.

# Materials and methods

## Experimental design

A total of 74 adult female Sprague-Dawley rats weighing 250–300 g were used in this study, divided in three sets of experiments: 54 for experiment 1 (behavioral assessment followed by motor mapping followed by cortical lesioning), 6 for experiment 2 (tract tracing) and 14 for experiment 3 (somatosensory maps). Animals were single housed and experiments were performed during the light cycle. All animal procedures were conducted in accordance with Drexel University Institutional Animal Care and Use Committee-approved protocols.

### Experiment 1

Animals received a complete spinal cord transection at thoracic level T8/T9 and were divided randomly into three groups: (1) animals treated with drugs and passive hindlimb bike exercise ('partial therapy', n = 15); (2) animals treated with drugs, passive hindlimb bike exercise and active treadmill training ('complete therapy', n = 15); (3) animal that receive sham drug (saline) and sham exercise therapy ('sham therapy', n = 15). Note that animals that received partial therapy also received sham-treadmill therapy. Hindlimb motor function was assessed as percentage of weight supported step cycles (%WSS) during treadmill locomotion (at 4, 8 and 12 weeks post transection), and in the open field using BBB scores (at 2, 4, 8 and 12 weeks post transection, normalized to group-averages at assess effects of therapy). One animal that underwent sham therapy died at week-12, so it was excluded from the analyses. Two animals under partial therapy had no treadmill data at week-8, so they were excluded from the %WSS analysis. During week-13 after the spinal transection, a subset of these animals were anesthetized and the motor cortex was mapped using intracortical micro-stimulation to identify changes in motor representations (complete therapy, n = 14; partial therapy, n = 12, sham therapy, n = 10). Reorganized areas were identified in this manner and bilaterally lesioned (complete therapy, n = 12; partial therapy, n = 10). All animals that received cortical lesions were given therapy for an additional 2 weeks and then were behavioral re-evaluated with %WSS and BBB scores. Two additional groups of intact naïve animals were used for control experiments (n = 5 for motor maps, n = 4 for cortical lesions).

### Experiment 2

Animals received a complete spinal cord transection at thoracic level (T8/T9) and either complete therapy (n = 2) or sham therapy (n = 2). Anterograde tract tracing was performed by injecting a tracer into the reorganized deafferented hindlimb motor cortex 9 weeks after the spinal transection. Animals continued therapy (complete or sham) and were perfused 3 weeks later. Axons were counted in the thoracic gray matter of the spinal hemicord above the level of the lesion, contralateral to the injected cortex. A group of naïve intact animals (n = 2) was used as a control.

### Experiment 3

Animals were chronically and bilaterally implanted with 4 × 4 16-channels arrays of microelectrodes in the hindlimb somatosensory cortex, were then spinally transected and received either complete therapy (n = 8) or sham therapy (n = 6). Somatosensory maps of the responses of hindlimb neurons to forelimb stimuli were performed in anesthetized conditions before (week 0) and 4, 8, and 12 weeks after spinal cord transection. Note that five animals were excluded from the study between week-8 and week-12 due to experimental complications (loss of head-cap, bladder infection or severe skin lesions). 11 animals were also recorded during treadmill locomotion.

## Spinal cord transection and animal care

Complete mid-thoracic spinal cord transection was performed with similar methods as in our previous studies (*Knudsen et al., 2012*; *Manohar et al., 2012*; *Ganzer et al., 2013*; *Graziano et al., 2013*; *Foffani et al., 2016*; *Ganzer et al., 2016*). Animals were anesthetized with 5% isoflurane and 2 L/min of oxygen and maintained at 2–3% isoflurane with 1-liter oxygen for the duration of the surgery. A laminectomy at the T8/9 level exposed one spinal cord segment. A #10 scalpel blade was used to open the dura and pia mater and #11 scalpel blade was used to make the complete transection of the spinal cord. A fine-tipped microaspiration device was then used to remove 2–3 mm of

spinal cord. A collagen matrix, Vitrogen (Cohesion Technology, Encinitas, CA), was injected into the site of the transection. Following recovery from surgery, animals were given an IM injection of the antibiotic Pen-G and 5 ml of lactated ringer subcutaneously animals and returned to their home cages.

Animals were housed two per cage with highly absorbent Alpha-Dri bedding (Shepherd Specialty Papers Inc. Kalamazoo, MI) and cages were kept on warm water blankets. Animals were housed under a 12 hr light/dark cycle (lights on at 07:00) with ad libitum access to food and water. In addition, bladder care was given three times daily for 2 weeks or until bladder control was regained. At the sign of infection, the rats were given subcutaneous injections of Baytril (0.06 mg/kg) once a day for 7 days.

## Therapies

### Drug administration

Drug administration was similar to our previous studies (*Ganzer et al., 2013*, *2016*). 5-HT agonists were dissolved in sterile physiological saline. Quipazine was injected intraperitoneally and 8-OH-DPAT subcutaneously. Animals were injected once per day 5 days per week beginning 2 weeks after the spinal transection and continuing until week-12. The 2 week lag time post injury allowed time for 5-HT receptor upregulation in the spinal cord caudal to the lesion (*Kim et al., 2001*). Our chosen dose of drug was a combined injection of 0.125 mg/kg of quipazine (1 mg/ml) and 0.125 mg/kg of 8-OH-DPAT (1 mg/ml), as this dose has a maximal effect on cortical reorganization (*Ganzer et al., 2013*). Sham therapy animals received an equivalent volume of saline injections.

### Passive hindlimb bike exercise

Passive hindlimb bike exercise was similar to our previous studies (*Ganzer et al., 2013*; *Graziano et al., 2013*; *Foffani et al., 2016*; *Ganzer et al., 2016*). Animals received this exercise three times a week (Monday, Wednesday and Friday) starting 1 week after the spinal transection, using a custom built motor-driven cycling apparatus. Rats were suspended horizontally with their feet secured to the pedals. Cycling speed was maintained at 45 revolutions/min and each exercise bout consisted of two 30 min exercise periods with a 10 min rest period in between. Sham exercise consisted of placing the animals on the bicycles for 70 min while the pedals were stationary.

### Active treadmill training

Active treadmill training was an improved version of what we used in previous studies (*Kao et al., 2009*; *Foffani et al., 2016*), such that the ability of the rats to accept hindlimb loading was tested weekly (*Timoszyk et al., 2005*). Animals received treadmill training 10 min after the drugs were injected each day. Rats were placed in a cloth harness with a Velcro strap and attached to an arm of a device that provided lateral and vertical weight assist. The arm was attached to a spring on the other end, which could be extended by turning a knob (*Figure 1—figure supplement 2*). In order to vertically support the rat's weight the spring was extended until the moment of the forces acting on either ends of the arm were balanced. The device was calibrated to obtain the percentage of the rat's body weight supported. At the start of each week, vertical weight assist was ramped down. As the hindlimbs were loaded with more of the animal's weight, they typically started taking steps. At a certain level of weight assist, the animal was unable to bear its own weight and failed to step with its hindlimbs for more than three consecutive step cycles. The assisted vertical weight support was set right above this value, which was defined as load-bearing failure point (*Timoszyk et al., 2005*). Two weeks after the spinal transection, animals were not able to step with their hindlimbs, so the initial failure point was set at the assisted weight support that allowed animals to maintain a quadrupedal position on the treadmill, supporting part of their weight with the forelimbs. This initial failure point at week-2 corresponded to the weight of the animal, used to calibrate the device, less the amount of vertical support provided by the forelimbs on the treadmill, which we could not measure. Therefore, the assisted vertical weight support at failure point was expressed as a percentage of the initial failure point at week-2 (note that 100% assisted weight support is upper bounded by the weight of the animal). Failure point was evaluated in all animals (i.e. also in animals that received sham therapy or partial therapy and thus did not receive treadmill training).

## Behavioral assessment

Behavioral measures in all groups were always performed on drugs (i.e. after acute administration of the same 5-HT agonists used for therapy), in order to guarantee the functional state of the cord below the level of the lesion (*Barbeau and Rossignol, 1990*; *Jackson and White, 1990*). We did not use tail pinch or perineal stimulation.

### Treadmill testing (% weight-supported step cycles)

Treadmill testing was performed without vertical support for all therapy groups, after administration of drug. Testing sessions were video-taped to determine the number of step cycles for which the animal could support its own weight at weeks 4, 8 and 12 post spinal cord injury (SCI). The recovery of hindlimb locomotor function was assessed using the number of weight supported step cycles (WSS) taken by the rat when attached to the weight assist device and placed in a neutral position on a moving treadmill. At the neutral position, the height of the rat above the treadmill was adjusted till the forepaws made complete plantar contact with the surface, and the length of the spring was adjusted till the arm was level and provided no additional force on the rat in the vertical direction (i.e. no vertical assist). The %WSS was calculated as the number of weight supported step cycles over the first 100 step cycles taken. The start of a step cycle was determined from forepaw footfalls. A cycle was defined as weight supported if for each hindpaw step the following criteria were fulfilled: (a) the hindquarters were elevated off the treadmill surface, (b) the hindpaw was placed underneath with visible muscle contraction, (c) the knee did not touch the treadmill belt, (d) the plantar surface of the hindpaw made contact with the treadmill during lift off and (e) again reestablished after the hindlimb was advanced. The experimenter performing the assessment from the videotape was blind to the group of the animal. During treadmill testing, animals did not receive any assisted vertical weight support, so the %WSS was used to directly measured the ability of animals to support their weight during hindlimb locomotion.

### Open field locomotion (open field score)

All therapy groups were evaluated for recovery of locomotor function during overground-locomotion using the Basso, Beattie and Bresnahan (BBB) open field score at weeks 2, 4, 8 and 12 post- SCI (*Basso et al., 1995*). The open field score, which was always estimated over 4 min, was normalized at weeks 4, 8 and 12 using the average score at week two for each group (after verifying that the was no difference among groups at week 2). During each evaluation week, the animals were tested 5 min after drug administration. Sham therapy animals received a challenge dose of drug prior to BBB testing. BBB scores from 0 to 8 are non-weight supporting, while scores of 9 to 21 indicate hindquarter weight support. Three experimenters performed the evaluation, one placed the animal in the open field and the other two were blind to the group of the animal and performed the scoring.

## Motor mapping

Motor mapping was performed during week-13 after the spinal transection using intracortical microstimulation with methods similar to our previous study (*Ganzer et al., 2016*). Animals were anesthetized with a combination of Ketamine (50 mg/kg), Xylazine (5 mg/kg) and Acepromazine (0.75 mg/kg) and placed in a stereotaxic frame. Supplemental doses of anesthetic were administered as needed. Craniotomies were performed over the right cortex to expose the medial post-bregma area. Electrode penetrations were defined using stereotaxic coordinates (*Leergaard et al., 2004*) on a cortical grid (coordinates AP: −0.5 to 2 mm posterior to bregma; ML: 1 to 3.5 lateral to midline). Electrode penetrations were made at 500–600 μm intervals within the medial post-bregma area. Care was taken to avoid surface cortical vasculature during mapping. A low impedance glass-insulated tungsten microelectrode (500 kΩ; FHC Inc.; Bowdoin, ME) was mounted on a stereotaxic electrode manipulator. In order to assess microstimulation waveform quality, a 100 kΩ resistor was connected to a grounding screw adjacent to the craniotomies in series with the stimulator (AM systems; Sequim, WA). The dura was removed and the microelectrode was lowered, perpendicular to the surface of the brain, to penetrate the pia. The microelectrode was then slowly inserted into the brain to a depth of ~1,600 μm, corresponding to cortical layer V/VI. Stimulation parameters consisted of 0.2 ms duration constant current bipolar pulses (anodal leading), at 333 Hz in trains of 300

ms duration. At each penetration site, the stimulation current gradually increased from 0 µA until a movement was evoked (current threshold). Movement was assessed by visual inspection. Penetrations were marked as non-responsive when no movement could be evoked with stimulation up to 100 µA. Motor maps were conducted with two experimenters. The first experimenter placed the electrode and recorded the data for each site. The second experimenter was kept blind to the experimental group of the animal and electrode placement to avoid potential biasing. The second experimenter delivered stimuli while observing which parts of the body moved in response to stimulation.

The expansion of intact motor representations into the deafferented hindlimb motor cortex was assessed by calculating the area (mm2) devoted to a movement type (trunk, hindlimb, forelimb, vibrissae) using custom MATLAB algorithms. Small shifts in this grid to avoid surface cortical vasculature during mapping were corrected for during processing. For each animal, the area of specific representations (trunk, hindlimb, forelimb, vibrissae) was determined by multiplying the number of responsive sites evoking the corresponding movement type by 0.25 mm2, using the method of (*Ramanathan et al., 2006*).

## Electrolytic cortical lesions

At the end of the motor mapping procedure, the reorganized area was bilaterally lesioned. Twisted pair stainless steel electrodes of 0.35 mm diameter insulated except at the tip was lowered to a depth of 1.65 mm below the surface of the brain at a stereotaxic coordinate that is 2 mm lateral to the midline and 1 mm posterior to bregma. A constant current of amplitude 500µA for 10 s was sent through the bipolar leads of the electrode. The electrodes were removed and the surface of the brain was covered with gelfoam and dental acrylic. Lesions were also performed in the same stereotactic location for naïve intact animals.

## Anterograde axonal tract tracing

Anterograde tract tracing was performed 9 weeks after the spinal transection. Rats were anesthetized and fixed in a stereotactic frame. A craniotomy was performed over the medial post-bregma area in the right hemisphere (0–2 mm posterior and 1–3 mm lateral to bregma). Six injections of the anterograde tracer, fluorescein-biotin-dextran (10% in PBS, 10 kDa molecular weight BDA, Mini Emerald, Invitrogen) was made to cover the de-afferented hindlimb region. All injections were at a depth of 1.5 mm below the surface of the cortex. Small holes in the dura were made before inserting the 33 gauge, 10 µl syringe (Hamilton Co.) controlled by an electrical pump (World Precision Instruments). The needle was left in place 5 min after insertion, followed by injection of tracer and again for 10 min after each injection to prevent backflow of tracer. Each injection consisted of 400 nl of tracer at a flow rate of 15 nl/min. After the last injection, the brain was covered with gelfoam and dental acrylic. Animals were returned to their therapy regime (sham, partial or complete therapy) and the tracer was allowed to be transported for 3 weeks before the animal was perfused at the end of 12 weeks after the spinal transection.

Animals were perfused transcardially with buffered saline (100 ml), followed by buffered 4% paraformaldehyde in 0.2 M PBS (500 ml). The thoracic cord (T1-T8) was removed, retaining the T1, T4 and T7 roots and post-fixed in 4% paraformaldehyde for 24 hr and finally cryoprotected in 30% sucrose until the tissue sank. Specimens were frozen in Shandon M-1 Embedding Matrix compound (Thermo Scientific) and sectioned on a freezing microtome at 25 µm. Labeled axons exiting the corticospinal tract were counted from five slices for each animal around T1, T4 and T7 similar to previous studies (*Carmel et al., 2010*; *van den Brand et al., 2012*). Slices used for counting were spaced at least 250 microns apart. For each slice, a region of interest was selected that included 2/3 of the spinal hemicord gray matter contralateral to the injected cortex, excluding only the most dorsal aspects of the dorsal horn and the most ventral aspect of the ventral horn (no labeled axons were found in these regions for any animal group). Stereo Investigator stereology program (MBF Bioscience) was used to sample the grey matter and count axons. Because cortico-spinal tract BDA-labeled terminations were sparse, we used relatively dense sampling parameters: square counting frames 50 µm on each side, resulting in sampling 50% of the region of interest per section. Data are reported as cumulative axon counts per area per slice. A group of naïve intact animals (n = 2) was used as a control.

## Somatosensory mapping

Somatosensory mapping was performed with similar techniques as in our previous studies (*Foffani et al., 2004*; *Tutunculer et al., 2006*; *Moxon et al., 2008*; *Kao et al., 2011*; *Knudsen et al., 2012*; *Manohar et al., 2012*; *Knudsen et al., 2014*). Microelectrode arrays were chronically implanted under general anesthesia (2–3% isoflurane in O2 delivered via orotracheal intubation) and aseptic conditions. More specifically, 4 × 4 arrays of 50 μm Teflon-insulated, stainless steel microwires (MicroProbes for Life Sciences) were bilaterally implanted to the infragranular layers (1.3–1.5 mm) of the rat hindlimb representation within the sensorimotor cortex (*Leergaard et al., 2004*).

On the days of somatosensory mapping, animals were lightly anesthetized with sodium pentobarbital (Nembutal) with an induction dosage of 35 mg/kg and maintained at stage III-2 level of anesthesia (*Friedberg et al., 1999*; *Erchova et al., 2002*). They were responsive to toe pinch and corneal reflexes. Seven sparse locations were stimulated on each forelimb including two digits, two palm pads on the forepaw and wrist, elbow and shoulder. Each location was consecutively tapped 100 times at 0.5 Hz with a fine-tipped metal probe, controlled by a precision stepper motor (Gemini GV6). To ensure that only tactile receptors at the sight of contact were activated and to control the magnitude of the stimuli at each location, the metal probe was first positioned on the skin, ensuring contact but no visual indentation under 10X magnification. The metal probe was then moved 0.5 mm away from the skin, and the stimulation was started. The effect of the stimulus was viewed under 10X magnification to ensure no movement of the digits or limb. All locations were tapped within the same recording session to ensure that the same neurons were recorded in response to stimulation of all locations. All 100 stimuli were given to a location, and then the stimulator was moved to the next location. The motor stimulator simultaneously sent a TTL-pulse to the data acquisition system to record the timestamp of the stimulus onset. Single neurons were discriminated by hand using a combined thresholding and real-time PCA analysis of waveform features. Neurons were re-discriminated before every recording session to ensure only single units were recorded. Neural signals were acquired (40 kHz) with a neurophysiological recording system (Plexon). The waveforms and action potential timestamps of all the discriminated neurons were recorded, and stored for further analysis.

The responses of neurons to sensory stimuli were quantified using peri-stimulus time histograms (PSTHs,[*Tutunculer et al., 2006*]). A PSTH is built as a histogram of spike times relative to the sensory stimulus (time-zero) averaged across trials (100 trials, 1 ms binsize). A neural response was considered significant if (a) it exceeded a threshold set as the average background activity of the neuron (evaluated from 100 to 5 ms before the stimulus) plus three standard deviations, (b) at least three bins (at 1 ms binsize) were over the threshold, and (c) the spiking activity between the first and the last significant bin was significantly greater than the background activity (non-paired t-test, $p < 0.001$). Response magnitude was quantified as the background-subtracted average number of spikes per stimulus in the 5–50 ms post-stimulus window of the PSTH.

## Awake recordings

Neuronal activity was also recorded while animals locomoted on the treadmill (n = 11). Chronically implanted electrodes were connected to headstages and tethered to the Plexon recording system. Neurons were discriminated before each recording session. A mirror was placed behind the animal and the entire session was videotaped in order to extract the timestamps of paw placements. The recording lasted 10 mins. The timestamps on the video recording were synchronized with the neural recording system.

PSTHs were generated around the forepaw footfalls using timestamps from the video analysis of treadmill locomotion using a 10 ms bin size. Background firing activity of each neuron was calculated as the firing rate in the entire recording duration. The PSTHs were smoothed in a 500 ms window (250 ms before and after paw placement) using a sliding-window zero-phase filter of length five bins. To see if the neuron was responsive to paw placements, a threshold was defined as the 99% confidence interval above the mean firing rate of the cell. The peak of the response was the highest bin that crossed the threshold and the response was defined as all the consecutive bins around the peak bin that were over the threshold. The response of the cell was calculated between these bins from the unsmoothed PSTH. Responses were considered significant if at at least three consecutive bins exceeded the 99% confidence interval threshold.

## Perfusion and histology of the spinal cord

Animals for Experiments 1 and 3 were perfused transcardially with buffered saline (100 ml), followed by buffered 4% paraformaldehyde in 0.2 M PBS (500 ml). For animals that underwent cortical lesioning, the brain and spinal cord were then removed, post-fixed in the same buffered 4% paraformaldehyde for 24 hr and finally cryoprotected in 30% sucrose until the tissue sank. Specimens were frozen in Shandon M-1 Embedding Matrix compound (Thermo Scientific, Waltham, MA) and sectioned on a freezing microtome at 40 µm. The transection segments of the spinal cords were sectioned horizontally into five sets. One set was Nissl–myelin stained and the resulting sections were examined under a microscope to confirm completeness of the transection. A second set was stained with a polyclonal antibody to 5-HT. Frozen sections were incubated at 4°C with the primary antibody (diluted 1:40,000 Immunostar, Stillwater, MN) for 16 hr, with biotinylated goat anti-rabbit IgG for 2 hr, and with avidin-biotinylated horseradish peroxidase complex for 2 hr, as specified by the manufacturer (ABC Standard Kit; Vector Laboratories, Burlingame, CA). Peroxidase reactivity was visualized with 0.05% diaminobenzidine tetrahydrochloride and 0.01% hydrogen peroxide in 0.05 mM Tris buffer.

## Statistical analysis

Statistical analyses were performed using two-way analyses of variance (ANOVA) or one-way multivariate ANOVA. A squared-root transformation was used when necessary to guarantee normality and homoscedasticity. Binary neural data (responding/ nonresponding neurons, sample size is number of neurons) were entered into generalized linear models (GZLM) with binomial distribution and logit link function. GZLMs allow binary data to be rigorously analyzed with ANOVA-like designs. Tukey's test or Fisher's test were used for post-hoc comparisons. Correlations were assessed with Pearson correlation coefficient and with robust regression methods, using a bi-square fit. All results were considered significant at $p < 0.05$.

## Acknowledgements

This work was supported by grant NS096971 by the National Institutes of Health,grant 1402984 by National Science Foundation and grant 85900 by the Shriner's Hospital for Children. Thanks to Gary Blumenthal, Chimela Nwaobasi and Timur Litvinov and for assistance with animal care, behavior and video analysis.

## Additional information

### Funding

| Funder | Grant reference number | Author |
|---|---|---|
| National Science Foundation | CBET 1402984 | Karen A Moxon |
| National Institutes of Health | NS096971 | Karen A Moxon |
| Shriners Hospital for Children | 85900 | Karen A Moxon |

The funders had no role in study design, data collection and interpretation, or the decision to submit the work for publication.

### Author contributions

AM, Conceptualization, Data curation, Software, Formal analysis, Validation, Investigation, Visualization, Methodology, Writing—original draft, Writing—review and editing; GF, Formal analysis, Supervision, Investigation, Visualization, Writing—original draft, Writing—review and editing; PDG, Formal analysis, Investigation, Methodology, Writing—review and editing; JRB, Supervision, Investigation, Writing—review and editing; KAM, Conceptualization, Resources, Supervision, Funding acquisition, Investigation, Visualization, Methodology, Project administration, Writing—review and editing

### Author ORCIDs

Anitha Manohar, http://orcid.org/0000-0002-3011-2623
Karen A Moxon, http://orcid.org/0000-0002-5790-097X

## Ethics

Animal experimentation: All animal procedures were conducted in accordance with Drexel University Institutional Animal Care and Use Committee-approved protocols (Protocol# 20069 and Protocol#18374)

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
