## [Decision Letter]

Thank you for submitting your article "Cortex-dependent recovery of hindlimb locomotion after complete spinal cord injury" for consideration by *eLife*. Your article has been reviewed by three peer reviewers, and the evaluation has been overseen by a Reviewing Editor and Eve Marder as the Senior Editor. The following individuals involved in review of your submission have agreed to reveal their identity: Arko Ghosh (Reviewer #1); Reggie Edgerton (Reviewer #3).

The reviewers have discussed the reviews with one another and the Reviewing Editor has drafted this decision to help you prepare a revised submission.

Summary

The present work investigates the neuronal plasticity following pharmacological and physical therapy leading to weight-supported hind limb stepping after complete spinal cord transection at the thoracic level in rats. The authors show that full weight-supported stepping is followed by plasticity in motor cortex that involves increased motor output to trunk muscles with a possible increased integration of sensory input to these areas from the forelimbs during training. The essential new idea is that improved locomotor function after a complete spinal cord transection can occur with interventions previously shown to result in improved locomotor function, but it is not via any crossing of fibers over the lesion but due to improved control of trunk muscles that act as a mechanical bridge to allow weight-supported stepping in the hind limbs. The study is comprehensive and the central hypothesis is substantiated by electrophysiological, lesion and anatomical findings. The study has the potential to provide important new insight to the functional recovery following intervention therapies after spinal cord injury. There are, however, a number of concerns that should be addressed to improve the presentation, make the conclusions of the paper stronger and clarify several issues.

Title: The title should include the animal used.

Essential revisions

1) A major concern raised by all reviewers is how the results are interpreted and discussed. The paper emphasizes throughout the observation that these animals, with appropriate rehabilitation and serotonergic administration, can recover function including weight supported stepping after complete spinal cord injury. Although this is an important point and bears repeating: it is not new. This result is well established in the literature, by several studies (from Rossignol, Courtine, Edgerton and others) in rats, cats, and mice using a variety of interventions. The point is still important to make as it is not always appreciated, but the paper has to be reworked throughout to avoid repeated overstatements/overselling with regard to these findings and by including appropriate acknowledgement and reference to previous work. On the other hand, the authors should focus in the writing on the real significance of the study: namely that cortical plasticity might mediate functional recovery after SCI via biomechanical coupling across the lesion – which is not well appreciated in the field. Also here the wording should be appropriate not overstate the finding.

2) The authors need to substantiate that they obtain WSS after training. All analysis for WSS is expressed as% of 100 steps that met the criteria of a step cycle on the treadmill and the criteria in the open field. These data therefore provide no direct evidence of absolute levels of weight support generated. All measures are% of the number of steps with the criteria for a step adjusted to the level of a constant criteria for treadmill and over-ground open field as well as for training. No absolute values even for the BBB tests are provided. BBB scores are informative as to the functional level of each animal and the scale is clearly not linear; an increase from 2 to 3 might mean something very different from an increase from 10 to 11. All changes are in% for each rat with adjusted levels of support as the point of reference. The subjective scores should be reported – so as to demonstrate that each group had similar functional levels after spinal cord injury.

3) The biomechanical coupling across the lesion is the core idea put forward in the study. Yet, there is little explanation or mechanistic insights into how this is accomplished. It will be useful to have a description of what animals actually did to recover weight bearing stepping. How different are the trunk-supported steps post-lesion from the steps pre-lesion? How did animals with good recovery control their trunk muscles differently than animals with poor recovery? The authors should be able to extract biomechanical data from video analysis of animals without further experimentation to address these questions. The open field-scores were estimated over the first 100 steps. Did the animals in all of the groups perform the tests as quickly and was this used to adjust for the corresponding statistics. It will also be useful to discuss possible mechanisms for the recovery across the lesion. There have been reports of intersegmental reflexes among trunk muscles that should be cited. Altogether the authors should better separate their work from previous research with a more in-depth consideration of what has been already done regarding trunk supported steps and the proposed mechanisms.

4) Changes in cortical plasticity. More excitable vs. more area? The authors argue the area occupied by the trunk region expands following the lesion and supports the treatment-induced recovery. However, it remains possible that the cortex is more excitable post-lesion – as suggested by some of the previous work performed by some of the co-authors of this very report. To resolve this, the data from the different threshold currents used in the ICMS experiments could be simply shown. Essentially, a heat map showing the current thresholds should settle this.

5) The cortical lesion experiments are incomplete unless the functional specificity of the lesion is clarified. The lesion experiments bear a strong importance to the study so it is important to address if the impact of the lesion is specific to the hind limb. This would be expected from the apparently accurate lesion location; but a control measure used to explore if the forelimb weight support remains unobstructed should be presented to exclude a general impact of cortical lesion (brain swelling etc.). In particular the authors should pay attention to the cortical lesion on normal locomotor behavior in non-spinal-cord-injured animals. This relates to their findings that cortical lesions in control animals have no effect on motor behavior. At the moment this is just mentioned in one sentence and would need some sort of qualification and discussed in relationship to previous findings in rats (e.g. Ölvecky's or Courtin's).

6) The sensory cortical recordings should be better aligned with the rest of the paper to support the central claim by adding trunk sensory simulations or at least discuss their role in the recovery. Secondly, in Figure 5 the 'quarter' division of the grid seems a bit arbitrary. It might be more informative to divide sites according to their motor representation. I.e. is the increased forelimb representation mainly at sites that encode the trunk or is it general across all representations? Finally, was there a phase relationship between the activity of these cells and forelimb movements during normal walking that could be explained by their new receptive field? This latter analysis of neural coding is beyond the scope of this paper, but it would be helpful if the authors could comment on it since it would clearly help in explaining the recovery strategy.

7) While the measurements of the protein levels for a number of molecules linked to plastic changes are interesting they presence or absence cannot be linked to any specific plasticity. This part of the results should be deleted from the study.

8) The Discussion needs to be reworked to take into account the new data analysis and to tone down the strong statements in the first paragraph. It should also include a discussion of other studies that have shown cortical plasticity after spinal cord injury and the effect of cortical lesions in intact animals.

9) The logic of tying this work to a "brain machine interface" seemed off track and should be avoided.

[Editors' note: further revisions were requested prior to acceptance, as described below.]

Thank you for resubmitting your work entitled "Cortex-dependent recovery of unassisted hindlimb locomotion after complete spinal cord injury in adult rats" for further consideration at *eLife*. Your revised article has been favorably evaluated by Eve Marder (Senior editor), a Reviewing editor, and three reviewers.

The manuscript has been improved but there are some remaining issues that need to be addressed before acceptance, as outlined below:

This revised study addresses most of the concerns raised by the reviewers. It has clarified issues regarding the biomechanical coupling and unassisted weight support which further helps to highlight the novelty of the study. The major strength of this report is in its focus on the circuits that are above the injury which may biomechanically influence the motor functions originally controlled by the injured cord. Although there is not a clear mechanistic explanation for the link between behavioral recovery, changes in cortical structures and the mechanical coupling the authors have gone a long way in the revision to explain how animals achieve training+treatment induced recovery and discussed the findings appropriately.

There remain some questions that the authors needs to respond to before the paper can be published:

1) There are significant concerns about the final figure showing the Western blots. Comments about this were made by reviewers in the previous round but it did not make it into the decision letter. First, several β actin control blots are reused across figures. I.e. in Figure B, the β actin control for PSD95 is the same as the control for synaptophysin. The authors should clarify that they were all taken from the same sheet. If they're not, then it's not appropriate to use the same β actin blot as a control for both cases. More confusing, however, are the blots in Figure C and D. The β actin controls appear to be the same for GLAST and ephrin A3, except that the final sample is missing in D. As for Figure B, if they were all on the same sheet, then this might be ok. But it appears to be more complicated. In C there are 5 samples for partial and 5 samples for complete; in D there are 4 samples for partial and 5 for complete. This explains why there's one less sample in the β actin control in D. However, this implies that there is a mismatch between columns in C and D, with the 5 complete animals matching up with different columns in the β actin controls (matching with 10-14 in C and with 9-13 in D). One can also see part of a blot clipped at the end of the ephrinA3 row. Please explain 1) whether the reused β actin controls were obtained from the sheet, 2) why there are 4 partial animals in some samples and 5 partials in others, 3) the apparent mismatch between columns in C and D, and 4) the apparently clipped blot in D.

Added by Senior Editor: it would be best if independent controls were used for each run. If they were run on one gel, then redesign the figure to make that clear and only show the control once if it was the same actual run used for multiple samples. fix this in some scientifically unassailable way or we won't be able to proceed to publication. Incorrectly replicated gel panels are one of the most common errors found in published papers, and we are starting to monitor papers for this kind of issue.

2) Please add the results from the bi-square vs. OLS regressions as a supplementary figure, along with your argument as to why the 'OLS' is preferred.

3) There is an opportunity to refer to a paper about intersegmental reflexes:

Reflexes evoked in human erector spinae muscles by tapping during voluntary activity. Tani T1, Yamamoto H, Ichimiya M, Kimura J.

---

## [Author Response]

*Title: The title should include the animal used.*

The title was changed to “Cortex-dependent recovery of unassisted hindlimb locomotion after complete spinal cord injury in adult rats”

*Essential revisions*

*1) A major concern raised by all reviewers is how the results are interpreted and discussed. The paper emphasizes throughout the observation that these animals, with appropriate rehabilitation and serotonergic administration, can recover function including weight supported stepping after complete spinal cord injury. Although this is an important point and bears repeating: it is not new. This result is well established in the literature, by several studies (from Rossignol, Courtine, Edgerton and others) in rats, cats, and mice using a variety of interventions. The point is still important to make as it is not always appreciated, but the paper has to be reworked throughout to avoid repeated overstatements/overselling with regard to these findings and by including appropriate acknowledgement and reference to previous work. On the other hand, the authors should focus in the writing on the real significance of the study: namely that cortical plasticity might mediate functional recovery after SCI via biomechanical coupling across the lesion – which is not well appreciated in the field. Also here the wording should be appropriate not overstate the finding.*

We agree with the reviewers that several previous studies were able to improve hindlimb locomotion after complete spinal cord injury in adult rats, either using sensorimotor rehabilitation (de Leon et al., 2002; Allouin, Delivet-Mongrain and Rossignol, 2015), administration of serotonergic agonists (Feraboli-Lohnherr, Barthe and Orsal, 1999; Antri, Orsal and Barthe, 2002; Antri, Barthe, Mouffle and Orsal, 2005), transplants below the level of the lesion (Jimenez y Ribotta et al., J Neurosci 2000; Sławińska et al., 2013) or combined pharmacological and electrical stimulation (Gerasimenko et al., 2007; Ichiyama et al., 2008; Courtine et al., 2009; Musienko et al., 2011). However, all previous interventions required either tail pinching, perineal stimulation and/or some level of assisted vertical weight support to achieve hindlimb locomotion in adult spinalized rats. In our study, during treadmill testing and BBB evaluation our animals received no form of peripheral stimulation or vertical weight support. This absence of external aid is important per se and it strengthens the finding that cortical plasticity might mediate functional recovery after spinal cord injury via biomechanical coupling across the lesion. This was clarified in the Discussion of the revised manuscript.

*2) The authors need to substantiate that they obtain WSS after training. All analysis for WSS is expressed as% of 100 steps that met the criteria of a step cycle on the treadmill and the criteria in the open field. These data therefore provide no direct evidence of absolute levels of weight support generated. All measures are% of the number of steps with the criteria for a step adjusted to the level of a constant criteria for treadmill and over-ground open field as well as for training.*

During testing, we did not provide any assisted vertical weight support to the animals, either on the treadmill or in the open field. The% WSS – which was assessed on the treadmill – thus reflects the ability gained by the animals to support their weight during hindlimb locomotion. This was clarified in the Materials and methods and Results.

During training, we did provide assisted vertical weight support to the animals through an external device that was controlled based on the animal’s ability to take steps (Timoszyk et al., 2005). At some level of weight support the animal is unable to bear its own weight and fails to step with its hindlimbs for more than 3 consecutive step cycles. The assisted weight support was set right above this value, which was defined as load-bearing failure point (Timoszyk et al., 2005). Two weeks after the spinal transection, animals were not able to step with their hindlimbs, so the initial failure point was set at the assisted weight support that allowed animals to maintain a quadrupedal position on the treadmill, supporting part of their weight with the forelimbs. This initial failure point at week-2 corresponded to the weight of the animal, used to calibrate the device, less the amount of vertical support provided by the forelimbs on the treadmill, which we could not measure. Therefore, the assisted weight support at failure point was expressed as a percentage of the initial failure point at week-2 (note that 100% assisted weight support is upper bounded by the weight of the animal). The failure point was evaluated in all animals (i.e. also in animals that received sham or partial therapy and thus did not receive treadmill training). The assisted weight support at failure point decreased with partial and complete therapy as a function of time after the spinal transection, providing direct evidence of increasing levels of weight support achieved by the animals. This was clarified in the Materials and methods and Results. We also added a figure (Figure 1—figure supplement 1) to show the evolution of the failure point after the spinal transection.

*No absolute values even for the BBB tests are provided. BBB scores are informative as to the functional level of each animal and the scale is clearly not linear; an increase from 2 to 3 might mean something very different from an increase from 10 to 11. All changes are in% for each rat with adjusted levels of support as the point of reference. The subjective scores should be reported – so as to demonstrate that each group had similar functional levels after spinal cord injury.*

Following the reviewers’ suggestions, we added in the Results the absolute values of the BBB scores at the beginning of therapy (i.e. 2 weeks after spinal cord injury): 6.5 ± 1.3 for sham, 5.7 ± 2.0 for passive, 5.2 ± 2.4 for full. All groups had similar functional levels at the beginning of therapy (1-way ANOVA, F(2,41)=1.7, p=0.19). Please note that, as stated in the original Materials and methods, these BBB scores were obtained on drugs (i.e. after acute administration of the same 5-HT agonists used for therapy).

*3) The biomechanical coupling across the lesion is the core idea put forward in the study. Yet, there is little explanation or mechanistic insights into how this is accomplished. It will be useful to have a description of what animals actually did to recover weight bearing stepping. How different are the trunk-supported steps post-lesion from the steps pre-lesion? How did animals with good recovery control their trunk muscles differently than animals with poor recovery? The authors should be able to extract biomechanical data from video analysis of animals without further experimentation to address these questions. The open field-scores were estimated over the first 100 steps. Did the animals in all of the groups perform the tests as quickly and was this used to adjust for the corresponding statistics. It will also be useful to discuss possible mechanisms for the recovery across the lesion. There have been reports of intersegmental reflexes among trunk muscles that should be cited. Altogether the authors should better separate their work from previous research with a more in-depth consideration of what has been already done regarding trunk supported steps and the proposed mechanisms.*

Our videos do not allow us to perform formal biomechanical analyses. Moreover, the animals were in a harness so we could not extract biomechanical data from the trunk. However, we did perform new video analyses from the treadmill testing data in animals that subsequently received cortical lesion (complete therapy, n=12; partial therapy, n=10), during testing at week-12 (pre-cortical lesion) and at week-14 (post-cortical lesion). Each of the 100 step cycles used for the% WSS was categorized – separately for each hindpaw – as no step (i.e. dragging), dorsal step (when the dorsal surface of the paw makes contact with the treadmill during paw placement), lateral step (when the animal performs a lateral sweep like motion without plantar contact) or plantar step (when the plantar surface of the paw comes in contact with the treadmill entirely during paw placement). Animals were categorized in two groups based on an arbitrary recovery threshold of 10% WSS at week-12: animals with good recovery (performer,% WSS>10, average 28.4 ± 11.1%, n=11) and animals with poor recovery (non-performer,% WSS≤10, average 2.8 ± 3.8%, n=11). Left and right hindpaw steps were separated as an independent factor. As expected, the percentage of no steps was lower in performers than in non-performers (3-way ANOVA, group: F(1,40)=10.1, p=0.0028) and increased after cortical lesion in both groups (lesion: F(1,40)=6.8, p=0.0129). The percentage of dorsal steps was similar in the two groups (group: F(1,40)=0.05, p=0.82), with a tendency to increase after cortical lesion in performers but not in non-performers (group x lesion: F(1,40)=5.3, p=0.0261). The percentage of lateral steps was again similar in the two groups (group: F(1,40)=1.2, p=0.28) and increased after cortical lesion in both groups (lesion: F(1,40)=6.8, p=0.0127). Finally, the percentage of plantar steps was markedly higher in performers (group: F(1,40)=95.7, p<0.0001) and significantly decreased after the lesion in performers (lesion x group: F(1,40)=19.9, p<0.0001; Tukey: p=0.0002) but not in non-performers (Tukey: p=0.11). These results clarify that the ability to make plantar contact with the treadmill during stepping is key to achieve good recovery and that the reorganized motor cortex plays a critical role in this recovery. This was added in a new figure (Figure 2—figure supplement 1)

Open field scores (i.e. BBB) were not estimated over these first 100 steps on the treadmill The scores were determined from a separate observation of the animals in the open field over a 4 minute period, according to standard procedures. This was clarified in the Materials and methods.

We could not find convincing references on intersegmental reflexes among trunk muscles. The only references we could find on intersegmental reflexes are related to respiration (McBain et al., Exp Physiol 2016 and references therein), which do not seems particularly relevant. Propiospinal connections (see e.g. Courtine et al., 2008; Filli et al., 2014) do not seem relevant either due to the completeness of our spinal cord transection. Suggestions are very welcome.

Unassisted trunk-supported steps have been described in detail in neonatally spinalized rats (Giszter, Davies and Graziani, 2007; Giszter, Davies and Ramakrishnan, 2008; Giszter, Hockensmith, Ramakirshnan and Idoekwere, 2010). A critical novelty of the present work is that we were able to achieve unassisted trunk-supported steps in adult spinalized rats. This was clarified in the Discussion.

*4) Changes in cortical plasticity. More excitable vs. more area? The authors argue the area occupied by the trunk region expands following the lesion and supports the treatment-induced recovery. However, it remains possible that the cortex is more excitable post-lesion – as suggested by some of the previous work performed by some of the co-authors of this very report. To resolve this, the data from the different threshold currents used in the ICMS experiments could be simply shown. Essentially, a heat map showing the current thresholds should settle this.*

As suggested by the reviewer, we included a figure (Figure 1—figure supplement 2) with the heat maps of the threshold currents used to evoke trunk movements for all three groups. There were no differences between the threshold currents across the three groups (1-way ANOVA, F(2,85)=0.751, p=0.475). Note that penetrations, rather than animals, were considered as individual samples for this analysis to maximize power. The heat maps indicate an expansion of the trunk area, particularly for animals that received complete therapy, while using equivalent threshold currents across all groups. This was clarified in the Results. Also note that our previous works only showed increased excitability in the somatosensory system. We have never found increased excitability in the motor system (Ganzer et al. 2014; Ganzer, Manohar, Shumsky and Moxon et al. 2016)

*5) The cortical lesion experiments are incomplete unless the functional specificity of the lesion is clarified. The lesion experiments bear a strong importance to the study so it is important to address if the impact of the lesion is specific to the hind limb. This would be expected from the apparently accurate lesion location; but a control measure used to explore if the forelimb weight support remains unobstructed should be presented to exclude a general impact of cortical lesion (brain swelling etc.). In particular the authors should pay attention to the cortical lesion on normal locomotor behavior in non-spinal-cord-injured animals. This relates to their findings that cortical lesions in control animals have no effect on motor behavior. At the moment this is just mentioned in one sentence and would need some sort of qualification and discussed in relationship to previous findings in rats (e.g. Ölvecky's or Courtin's).*

We agree that the specificity of the cortical lesion to hindlimb function is important. The% WSS for the forelimbs was 100% for all animals at week-12 and remained 100% for all animals at week-14, after the cortical lesion. Furthermore, from the new video analyses we confirmed that 100% of forelimb steps were plantar steps both before and after the cortical lesion. Forelimb weight support thus remained unobstructed by the cortical lesion, supporting the specificity of the cortical lesion to hindlimb function. This was clarified in the Results.

With our measures of% WSS we could not find any difference in hindlimb locomotion after cortical lesion in naïve non-spinal-cord-injured rats (i.e. 100% WSS). Our cortical lesion was rather focal, but we cannot exclude more subtle kinematic changes, which could be previously observed after extensive lesions of the forelimb and hindlimb sensorimotor cortex in mice (Ueno and Yamashita, Exp Neurol 2011). Nevertheless, the negligible effects of our cortical lesion in naïve animals are in agreement with recent data suggesting that motor cortex is required for learning but not for executing a learned motor skill in normal rats (Kawai et al., 2015), supporting the view that in rodents the motor cortex is not required for normal locomotion (Courtine et al., 2007). Importantly, our data suggest that the motor cortex is indeed required to sustain the recovered locomotion after complete spinal cord injury. This was clarified in the Results and Discussion.

*6) The sensory cortical recordings should be better aligned with the rest of the paper to support the central claim by adding trunk sensory simulations or at least discuss their role in the recovery.*

Neurons recorded from the putative hindlimb sensory region did not have significant responses to trunk stimulation neither before nor after spinal cord injury. The likely reason is that the trunk sensory area is not topographically adjacent to the deafferented hindlimb region. In normal animals the trunk sensory region is further caudal and lateral, well separated from the motor region. The novel sensorimotor map in the cortex that develops after therapy shows an overlapping expansion of the trunk motor (not sensory) and forelimb sensory into the deafferent hindlimb cortex. This speaks towards sensorimotor integration of the sensation in the forepaws that cues enhanced motor control of trunk musculature as a rehabilation strategy used by the animals. This was clarified in the Results

*Secondly, in Figure 5 the 'quarter' division of the grid seems a bit arbitrary. It might be more informative to divide sites according to their motor representation. I.e. is the increased forelimb representation mainly at sites that encode the trunk or is it general across all representations?*

In Figure 5, the quarters have been named according to their identity as forelimb/hindlimb/trunk sensory or motor representations in normal animals (changed in figure)

*Finally, was there a phase relationship between the activity of these cells and forelimb movements during normal walking that could be explained by their new receptive field? This latter analysis of neural coding is beyond the scope of this paper, but it would be helpful if the authors could comment on it since it would clearly help in explaining the recovery strategy.*

We performed awake recordings from the hindlimb sensorimotor cortex during treadmill locomotion in 11 intact rats. We recorded an average of 33.8 ± 8.3 neurons per animal. Of these cells, 24.81 ± 11.56% significantly responded to contralateral forepaw placement, with a broad distribution of peak latencies (Figure 5—figure supplement 1). This suggests that neurons in the hindlimb sensorimotor cortex are responsive to all phases of normal locomotion, providing a functional substrate for the reorganization of this cortical region – with proper treatment – after spinal cord injury. This was added in the Materials and methods and Results.

*7) While the measurements of the protein levels for a number of molecules linked to plastic changes are interesting they presence or absence cannot be linked to any specific plasticity. This part of the results should be deleted from the study.*

We now make clear that the measured proteins are related to synaptic plasticity and are specific to mechanisms of long-term potentiation (Filosa et al., Nat Neurosci 2009). Importantly, this allows our cortical reorganization to be linked to specific synaptic plasticity within the cortex and not just a response to subcortical changes. This was clarified in the Discussion.

*8) The Discussion needs to be reworked to take into account the new data analysis and to tone down the strong statements in the first paragraph. It should also include a discussion of other studies that have shown cortical plasticity after spinal cord injury and the effect of cortical lesions in intact animals.*

The discussion was reworked. See also response to comment 1.

*9) The logic of tying this work to a "brain machine interface" seemed off track and should be avoided.*

The wording “brain machine interfaces” was avoided.

[Editors' note: further revisions were requested prior to acceptance, as described below.]

*There remain some questions that the authors needs to respond to before the paper can be published:*

*1) There are significant concerns about the final figure showing the Western blots. Comments about this were made by reviewers in the previous round but it did not make it into the decision letter. First, several β actin control blots are reused across figures. I.e. in Figure B, the β actin control for PSD95 is the same as the control for synaptophysin. The authors should clarify that they were all taken from the same sheet. If they're not, then it's not appropriate to use the same β actin blot as a control for both cases. More confusing, however, are the blots in Figure C and D. The β actin controls appear to be the same for GLAST and ephrin A3, except that the final sample is missing in D. As for Figure B, if they were all on the same sheet, then this might be ok. But it appears to be more complicated. In C there are 5 samples for partial and 5 samples for complete; in D there are 4 samples for partial and 5 for complete. This explains why there's one less sample in the β actin control in D. However, this implies that there is a mismatch between columns in C and D, with the 5 complete animals matching up with different columns in the β actin controls (matching with 10-14 in C and with 9-13 in D). One can also see part of a blot clipped at the end of the ephrinA3 row. Please explain 1) whether the reused β actin controls were obtained from the sheet, 2) why there are 4 partial animals in some samples and 5 partials in others, 3) the apparent mismatch between columns in C and D, and 4) the apparently clipped blot in D.*

We agree with the reviewer’s original suggestion and have removed this figure (and two authors) from the paper.

*2) Please add the results from the bi-square vs. OLS regressions as a supplementary figure, along with your argument as to why the 'OLS' is preferred.*

Added these results as Figure 2—figure supplement 2, including further explanation with our argument for preferring OLS.

*3) There is an opportunity to refer to a paper about intersegmental reflexes:*

*Reflexes evoked in human erector spinae muscles by tapping during voluntary activity. Tani T1, Yamamoto H, Ichimiya M, Kimura J.*

We have now included this citation in the Discussion.